# Novel Endovascular Interventional Surgical Robotic System Based on Biomimetic Manipulation

**DOI:** 10.3390/mi13101587

**Published:** 2022-09-24

**Authors:** Chao Song, Shibo Xia, Hao Zhang, Lei Zhang, Xiaoye Li, Kundong Wang, Qingsheng Lu

**Affiliations:** 1Department of Vascular Surgery, Shanghai Changhai Hospital, Navy Medical University, Shanghai 200433, China; 2Department of Instrument Science and Engineering, Shanghai Jiao Tong University, Shanghai 200240, China

**Keywords:** medical robots and systems, surgical robotics, biomimetics, endovascular repair

## Abstract

Endovascular therapy has emerged as a crucial therapeutic method for treating vascular diseases. Endovascular surgical robots have been used to enhance endovascular therapy. However, to date, there are no universal endovascular surgical robots that support molds of different types of devices for treating vascular diseases. We developed a novel endovascular surgical robotic system that can independently navigate the intravascular region, advance and retract devices, and deploy stents. This robot has four features: (1) The bionic design of the robot can fully simulate the entire grasping process; (2) the V-shaped relay gripper waived the need to redesign special guidewires and catheters for continuous rotation; (3) the handles designed based on the feedback mechanism can simulate push resistance and reduce iatrogenic damage; and (4) the detachable design of the grippers can reduce cross-infection risk and medical costs. We verified its performance by demonstrating six different types of endovascular surgeries. Early evaluation of the novel endovascular robotic system demonstrated its practicability and safety in endovascular surgeries.

## 1. Introduction

Endovascular interventional therapy technology can be used to establish an operational channel in the vessel lumen through a working guidewire. Different types of interventional instruments are used to conduct a successful therapy [1]. The endovascular therapy is executed using X-ray images providing visual feedback, and the status of blood vessels is ensured by tactile feedback information obtained from the surgeons’ hands. Based on the operating habits, the endovascular techniques can be simplified into the following three combinations of motions: (1) push and retreat: the forward and backward movement of the intracavitary therapeutic apparatus in the blood vessels; (2) rotation: changes in the direction of intracavitary therapeutic apparatus in blood vessels; and (3) push and turn coordination: vessel navigation and lesion location are performed by simultaneously pushing and rotating the instruments [2]. However, interventional therapies require long working distances. The longest displacement distance of the handle can be 40 cm during the stent release [3]. In addition, during the arterial navigation, the guidewires and catheters require continuous rotation. The advancements in effective endovascular surgical robots have satisfied the requirements of long-distance movement along the long axis while rotating continuously. The maximum diameter of the aortic stent graft system is >40 mm [3]. The thinnest guidewire has a diameter of only 0.36 mm [4]. Thus, the chucking power of the endovascular robot should be effectively controlled and have a span of two orders of magnitude. The self-expanding stent graft system is primarily designed for releasing the stent between the outer sheath and inner core [5]. Thus, the two mechanical arms of the endovascular robot should move in the same or opposite directions. Different parts of the instrument perform different actions to complete sheath withdrawal and other actions. Although many devices operate along guide wires [6], rapid exchange devices do not move in the same direction as that of the wire. Thus, the robot should be designed such that it can meet the manipulators’ adjustment requirements along the short axial direction during surgery. This would maximize the simulation of the surgical process under the operation of the operator and assistant. That is, to meet the motion demands of the manipulators, the robotic system requires the short axial position to be changed during the procedure, which will maximize the simulation.

The Sensei X Robotic System, developed by Hansen Medical, controls the direction of the catheter tip via a mechanical pulling motion [7]. The Magellan robotic system, a second-generation robotic interventional surgical system, received a 510 (k) license from the U.S. Food and Drug Administration (FDA) in 2012 for its ability to manipulate guidewires and catheters during peripheral vascular interventional procedures [8]. Corpath 200 (Corindus Vascular Robotics, Inc.) can meet most of the work requirements of the 0.014-inch rapid exchange system, with catheters up to 7 French (Fr) in diameter [9]. Currently, it is the only endovascular surgical robot approved by the Food and Drug Administration and Conformitè Europëenne (CE) for coronary and peripheral interventional therapy. The PRECISE study confirmed the safety and feasibility of a robot-assisted percutaneous coronary intervention (PCI) via the Corpath system. Subsequently, the RAPID study aimed to prove the safety and feasibility of the treatment in peripheral artery lesions [10]. Robotic-assisted treatment was performed on a total of 29 lesions in 20 patients. The technical success, safety, and clinical success rates were 100%. However, remedial stents, which could only be used manually, were required in 34.5% of these cases. Other studies have confirmed that Corpath GRX can execute intracranial vascular coil embolization and thrombectomy stent release [11]. A clinical study (NCT04236856) is currently in progress to evaluate the efficacy of Corpath GRX in intracranial aneurysm embolization. Owing to the limitations in the operating distance of the system, the common carotid artery, which differed from the clinical applications, was selected for animal experiments. R-One, a new-generation coronary interventional robot developed by RoboCath, received CE certification in February 2019. In the field of interventional cardiology, R-One is the first CE-certified robotic system in Europe. The system design is similar to that of Corpath, with additional guidewire locking technology and a double-guidewire retention function. R-One™ has been marketed in Europe and Africa. The first tele-control robotic coronary intervention was performed in France in January 2021. However, analogous to Corpath, R-One is designed for rapid exchange systems. It contains two sets of non-coaxially distributed driving devices. It cannot satisfy the requirements for coaxial operation of over-the-wire (OTW) devices. The application of large OTW devices has significantly increased operating distance requirements, with clamping requirements for devices with large diameters. Moreover, this robot consists of several pairs of rollers. As a result, slippage between the treatment instrument and driving wheel is inevitable. Enhancing the clamping force to minimize the slippage may damage the surface of the catheter/guidewire, whereas a small clamping force may increase the slippage. Furthermore, it is expensive because the rollers are not reusable.

RobEnt, which is an endovascular surgical robot, was designed by Guo et al. [12]. A mechanical gripping design was adopted to minimize the uncertainty and damage associated with the rolling friction. A double manipulator design was also included. A fixed device was added to the front hand to ensure that the guidewire and catheter were fixed in one place when the back hand moved. In addition, the system has a rigid traction of steel wire to achieve guidewire and catheter coordination and meet the OTW equipment requirements. It is similar to the first-generation surgical robot initially studied by our research group [13,14]. However, the axial direction could not be altered when a rigid connection was adopted. Therefore, it cannot be applied to both the rapid-exchange (RX) and OTW systems. Moreover, because there are only two manipulators in the system, an outer sleeve is added to the transmission guide of RobEnt to avoid guidewire and catheter buckling during delivery. It limits the applicable diameters of endovascular devices and shortens the effective working distance between the guidewire and catheter. In summary, the brachial artery approach is not suitable for patients with lesions below the knee. The femoral artery approach is not suitable for patients with lesions above the arch.

Thus, a new type of surgical robot platform has emerged as a need to meet the requirements of both rapid exchange system appliances and OTW systems. This study improved on our previous studies by developing a new type of vascular interventional robot [14]. Its performance was verified experimentally in vitro and in vivo.

## 2. Materials and Methods

### 2.1. An Overview of the Novel Endovascular Interventional Surgical Robotic System

The novel endovascular interventional surgical robotic system mainly consists of two parts: a functional unit and interventional console (Figure 1). The functional unit adopts a biomimetic design, making it compatible with both RX and OTW catheter systems. The functional unit has two groups of mechanical arms with three degrees of freedom. Each arm has two manipulators, and each manipulator has one gripper. The grippers are detachable so that they can be replaced to accommodate varying endovascular devices. The interventional console is built outside the catheter lab, enabling the operator to perform surgeries remotely and avoid long-term radiation exposure. The manipulator can be switched through selection buttons and the surgeon uses four joysticks to manipulate the robotic system. The real-time state of the functional unit is displayed on the monitor inside the console. Moreover, fluoroscopic images, electrocardiograms, and hemodynamic information can be seen on the monitor at a close distance as well.

### 2.2. Functional Assessment of Bionic Designed Mechanical Arm of the Novel Endovascular Surgical Robot

The two robotic arms are arranged to mimic an operator–assistant stance, making it possible for large-length endovascular device delivery, such as multipurpose angiographic catheters (1.35 m), preventing pollution from end bouncing, and meeting the requirement of coaxial and non-coaxial manipulation. Each group of manipulators is a simulation of two hands of a surgeon. Each pair of grippers are like a thumb and an index finger. Manipulators and grippers can imitate the advancement, retraction, rotation and clamping of human hands and combinations of them.

As shown in Figure 2a, the operator and the assistant advance the stent graft system in a coaxial manner. As part of the novel vascular interventional surgical robotic system, two groups of independent mechanical arms were designed. These arms were capable of maximizing the simulation of this operational requirement, as shown in Figure 2b, with 18 degrees of freedom (DOF). Thus, the arms could independently complete the movement in three dimensions (*X*-axis, *Y*-axis, *Z*-axis) (Figure 2c).

The surgeon can control the long axial movement of the manipulators by operating the joystick with four grippers clamping the endovascular devices. The long axial movement of manipulators can be transformed into the movement of devices. The motion is directly driven by the three articulate modules. Kinematics and articulate module encoders were used to compute axial displacement. The desired long axial velocities were measured. The maximum speed error rate was 6.7%. In addition, we designed a proportional-integral-derivative (PID) controller to regulate the axial motion speed of the manipulator. This ensured that the controller could maintain a stable speed in case of an external interference. The advanced RISC machine (ARM) processor outputs the motor control signal. The encoder measures the axial motion speed. The expected speed was set at 13.75 cm/s, with sampling every 200 ms. When the system was stable, the bit error rate was 0.05. The basic performance parameters of the robot are as follows (Table 1):

These parameters met the technical standards required to introduce and release the aortic stent graft system. The new robotic system can complete surgical operations without intraoperative movements. Therefore, it shortened the operation time and reduced the risk of intraoperative contamination.

The movements of each manipulator in the same direction along the long axis can meet the delivery requirements of the guidewire and catheter. One or two manipulators can also be fixed upon request. The guidewire can be held by the manipulator. This minimizes the risk of damage to the vessel wall caused by guidewire movement after catheter advancement. The self-expanding stent release process can be executed by moving the manipulators in opposite directions. Fixation of the manipulators could effectively eliminate hand tremors and stent migration caused by relative movement of the hand. Therefore, it improved the success rate of the operation.

The independent up, down, left, and right movements of the four manipulators were adjusted via the control console to realize the four-manipulator coaxial state (Figure 3a,b). The joint participation of both the simulated operator and assistant in the operation enabled it to work. The four-manipulator coaxial state was suitable for the OTW-type devices.

The manipulators could also be adjusted to obtain a non-coaxial state (Figure 3c) by controlling the console, satisfying the requirements of rapid exchange devices and special operational requirements, including double-guidewire technology. It can also simulate the step-by-step operation process of the operator and assistant to the greatest extent, meeting the practical clinical application requirements.

### 2.3. Functional Assessment of the Detachable Self-Adaptive Progressive-Rotating Manipulator

The manipulator of the novel endovascular surgical robot consists of two direct-current (DC) motors, two encoders, and a step motor. The structure of the manipulator has been described previously in [14]. The manipulator was clamped and loosened using a DC motor (with 6 V voltage). An encoder was used to control the degree of clamping. The double V-shaped main grippers ensured that the center line of the endovascular instrument was always located at the center of the manipulator. This ensured the complete coaxial operation of the OTW-type instrument. In addition, the double V-shaped design could meet the clamping requirements from the finest 0.36 mm (0.014 inch) guidewire to the thickest 40 mm stent system (Figure 4).

The main V-shaped gripper can be divided into two modes: clamping and holding. Holding restricted the treatment device to the space between the double V-shaped grippers. However, it did not restrict its movement along the long axis. Thus, it avoided the pollution caused by the end bouncing during rotation. Clamping maintains the treatment device in a locked state. It completely follows the movement of the manipulator to push, retract, and rotate.

Advancement and rotation of the 5 Fr catheter were used to evaluate the manipulator. A dynamometer with 0.01 N accuracy was used to obtain the friction between the catheter and manipulator. The motor speed was measured using a software whereas the friction force at different speeds was measured using a dynamometer. In this system, the extent of catheter holding affected the driving motor speed. Clamping the catheter depended on the speed of the driving motor. Hence, we assessed the relationship between the motor speed and friction to evaluate the manipulator performance. This played a key role in improving the introduction of the endovascular instrument and rotation accuracy. The maximum friction force is 13.94 N based on [13]. This is similar to the manual operation force and satisfied the pushing demands of the treatment device.

The rotational motion of the manipulator was mediated by a stepping motor. The maximum and minimum rotation angles of each pulse were 61.2° and 0.18°, respectively. The maximum and minimum rotation speeds were 1.57 and 0.52 rad/s, respectively. However, during the visceral artery navigation, the guidewire and catheter must be rotated by 180° to determine the appropriate direction. Therefore, the function of continuous rotation must be equipped in the robotic system. A third group of assisted grippers was introduced to act as rotation relays. After reaching the edge, the manipulator can be automatically released/clamped. This represents the reception of rotation instructions and clockwise rotation. The V-shaped gripper was clamped when the manipulator reached its boundary. After the manipulator was released and rotated counterclockwise to the boundary, it was clamped again. The assisted gripper was released. After the manipulator was rotated clockwise to the boundary again, the above action was repeated to ensure a continuous rotation. A video of the grippers completing the rotation of the devices is provided in the Appendix A.

In addition, to match the shapes of the operation handles of different types of endovascular devices, the grippers of the manipulator were designed as detachable types (Figure 5), whereas V-shaped ends of different sizes could be selected according to the requirements.

### 2.4. Master-Slave Operation and Real-Time Performance Evaluation of the Novel Endovascular Surgical Robotic System

A master–slave operation design was adopted in the system, as shown in Figure 6. The design had three and six ARM processors at each core, respectively, in which distributed parallel control was performed. A CAN communication protocol was used between the master and slave. The system is equipped with two sets of master joysticks (Figure 1). Each joystick has a grip sense, which can rotate and advance simultaneously. Rotation was monitored using an encoder whereas advance and retreat were monitored using a grating-scale displacement sensor. The master hand is responsible for sending the instructions of movement via the CAN bus. The slave sets had two arms, each of which had two manipulators at the end. A single arm and manipulator could clamp, rotate, advance, and retreat the endovascular equipment, whereas the cooperation of the two arms could synchronously and independently operate the catheter and guidewire. The slave sets had 18 DOF and high flexibility, as shown in Figure 2b. It could recreate operational activities of most doctors. In terms of operational synchronization, communication delay and actuator response time were considered. A delay test was performed from the command of the operating joystick to the manipulator’s expected action. The delay of forward movement from the surgeon was calculated to be 336.0 μs. The delay of retracting, turning left, turning right, and clamping were calculated to be 385.63, 372.56, 371.77, and 449.70 μs, respectively. Although these communication delays were inevitable, the surgeon’s operational requirements were satisfied.

### 2.5. Functional Assessment of Force Feedback Handle of the Novel Endovascular Surgical Robot

To monitor the insertion resistance of the guidewire during intervention, the manipulator was equipped with a highly sensitive force sensor [15]. This resistance was transmitted through the CAN bus to the force feedback handle. It was responsible for generating a force on the surgeon’s hand so as to provide a feeling of resistance, as shown in Figure 7 [16]. The force feedback controller used PID algorithms. To evaluate the force feedback effect, 3D printing technology was used to reconstruct the vascular model. The abdominal aorta was approximately 13 mm in diameter. The common iliac artery was approximately 8 mm in diameter [17]. Tracking the push and pull motions during the master controller operation indicated that the feedback force could track the resistance up to 7 Hz. Compared with the measured resistance, the delay time of the tracking resistance was approximately 30–50 ms. Based on the bandwidth limitations of the handle, the feedback forces were smoother than the resistance. Owing to the difficulty in crossing the iliac artery to execute the crossover operation during endovascular surgery, cooperation between the guidewire and catheter was required. Therefore, we utilized an in vitro model to validate the practical application of the force feedback handles. Two slave manipulators were used to grab the guidewire and catheter. Guidewire resistance was perceived and fed back to the handle. Turning over the mountain required more force than non-turning over. The force feedback from the handle was tracked. The resistance was measured. The delay time was less than 60 ms. The peak error was 0.15 N. The root mean square (RMS) error was less than 0.05 N. Various catheter and guidewire specifications were tested in this model. When the catheter was changed from 6 to 9 Fr, the force was approximately equal to that of the 4 Fr. However, at bifurcation, the force increased from 6 to 9 Fr, with the maximum force of 12.5 N. The guidewire also behaved in a similar manner.

### 2.6. Animal Model

Ten male domestic swine (45 kg, 4 months old) were used for 38 cases of animal experiments in accordance with international regulations for the protection of laboratory animals. The experimental protocol was approved by the Ethics Committee of the Shanghai Changhai Hospital. The animals were anesthetized under general anesthesia. Vascular access was achieved through the femoral artery. A 6 Fr catheter was then inserted into the artery. At the end of catheterization, the animal was placed in the supine position on a table and was immobilized using straps to limit movement.

## 3. Results and Discussion

### 3.1. Assessment of the Novel Endovascular Surgical Robot in an In Vivo Model

Different target surgical procedures, including coronary artery stent implantation, self-dilated aortic covered stent implantation, balloon-expanded renal artery stent implantation, iliac artery stent implantation, and inferior vena cava filter implantation were selected to verify the application of the novel surgical robot.

The anesthetized animals were placed in the supine position on the operating table. Using the Seldinger technique, the surgeon punctured the right common femoral artery. An 8 Fr vessel sheath (Terumo, Tokyo, Japan) was introduced to establish the right femoral artery access. A 0.035 inch guidewire was inserted into the vessel sheath for preoperative preparation.

Five surgeons with varying experience took part in the operations. Two of them performed 500 operations a year. The other three performed 300, 100 and 10 operations a year, respectively. Before the operations, every surgeon was trained and familiarized with the robotic system. Except for the surgeon who performed 10 operations a year, who needed to be trained for 50 min, the others only needed 25 min of training. All surgeons were then qualified to use the robotic system.

#### 3.1.1. Percutaneous Coronary Intervention

A 5 Fr catheter (JL4) was inserted along the guidewire into the arterial sheath. Afterwards, four manipulators were positioned as follows: manipulator A held the proximal end of the catheter; manipulator B held the distal end of the catheter; manipulator C held the proximal end of the guidewire, and manipulator D held the distal end of the guidewire. All four manipulators were coaxial. Manipulator B clamped the catheter. Manipulator C released the guidewire and moved it back to the vicinity of Manipulator D as instructed by the surgeon. Subsequently, the guidewire was clamped by Manipulator C and pushed forward along the long axis to the vicinity of manipulator B. Then, the catheter was pushed forward along the direction of the guidewire by manipulator B to the vicinity of manipulator A and released. This was followed by moving manipulator A to an appropriate position on the side of the sheath. The procedure was repeated until the guidewire tip was directed towards the proximal part of the aortic arch. The guidewire was clamped using manipulator C. Manipulator B continuously sent the catheter forward to bend it against the non-coronary sinus edge. It was rotated until the tip of the catheter pointed towards the ostium of the left coronary artery. The joystick was slowly operated forward after adjusting the speed of the mechanical arm. Manipulator B was retracted until the catheter tip entered the ostium of the right coronary artery. Manipulator C/D retracted the guidewire into the sheath. The surgeon withdrew the guidewire and manually pushed the contrast agent to ensure that the catheter was in place. Subsequently, right coronary angiography was performed. The implantation site of the stent was established according to the imaging results. Manipulators C and D coordinatively sent a 0.014 inch guidewire to the right coronary artery. Based on the measurement results, a 3–20 mm balloon-expanded stent (Abbott, Lake County, IL, USA) was selected. The stent was introduced into the arterial sheath. Manipulators A and B clamped the proximal and distal ends of the stent system and manipulators C and D clamped the proximal and distal ends of the guidewire, respectively. They were at an angle of 30° with manipulators A and B. Manipulator D held the distal end of the guidewire. Manipulators A and C coordinated with the stent system to implant a balloon-expanded stent in the coronary artery. A detailed operation video is presented in Appendix A.

#### 3.1.2. Thoracic Endovascular Aortic Repair

A 5 Fr pigtail catheter (Cook Inc., Bloomington, IN, USA) was inserted along the guidewire into the aortic arch. Afterwards, the four manipulators were positioned as follows: Manipulator A held the proximal part of the guide catheter; manipulator B held the 5 Fr catheter end, and manipulators C and D held the proximal and distal parts of the guidewire, respectively. The four manipulators were coaxial. The surgeon substituted a 0.035 inch Amplatz guidewire (Boston Scientific, Marlborough, FL, USA). After manipulator C clamped the guidewire and sent it to the ascending aorta, manipulator D clamped and fixed it. Manipulators B and C coordinated to pull the pigtail catheter into the arterial sheath. The 24–80 mm stent graft (Microport, Shanghai, China) was selected based on the imaging measurement results. The surgeon then pulled out the arterial sheath and introduced the stent graft system into the femoral artery along the guidewire. Manipulator A clamped the proximal part of the system and manipulator B clamped its operating handle. Manipulator C was released, and manipulator D clamped the guidewire. Manipulators A and B were clamped alternately to direct the stent into a predetermined position. Manipulator A held the proximal part of the stent graft system. Manipulator B clamped the operating handle. Manipulator C clamped the inner core of the system. Manipulator D clamped the guidewire. Surgeon-controlled manipulator B retracted the operating handle to completely release the stent graft. Manipulator C gradually retracted the inner core. It then rotated moderately as it approached the proximal end of the stent. The friction between the system tip and the stent graft was avoided to avoid stent migration. A detailed operation video is provided in the Appendix A.

#### 3.1.3. Balloon-Expanded Stent Implantation of Renal Artery

A 6 Fr arterial sheath was introduced into the femoral artery after the femoral artery puncture. After introducing the guidewire, the four manipulators were positioned as follows. Manipulators A and B held the proximal and distal parts of the renal artery sheath, respectively. Manipulators C and D held the proximal and distal parts of the guidewire, respectively. All four hands were coaxial. As per the procedure in Section 3.1, a renal artery sheath was introduced at the renal artery level. The 5 Fr pigtail catheter (Cook Inc.) was used to perform abdominal aortic angiography. The 4 Fr intervention catheter (MPA-1, Cordis, Miami Lakes, FL, USA) was introduced into the arterial sheath. Subsequently, manipulator A clamped the arterial sheath. Manipulator B clamped the catheter. Manipulator C clamped the guidewire. Manipulator D held the guidewire. Manipulators B and C coordinated to send the catheter above the renal artery level. Manipulator C pushed and fixed the guidewire into the renal artery. After an angiography, the guidewire was changed to 0.014 inches. A 3–15 mm balloon-expanded stent (Firebird, Microport) was selected. Manipulator A clamped the proximal part of the stent system. Manipulator B clamped the distal end. Manipulators C and D clamped the distal end of the guidewire at an axial angle of 30° using manipulators A and B. The stent system was clamped alternately using manipulators A and B. The stent was directed to a predetermined position. Eventually, the surgeon manually released the stent. Manipulators B and D held the distal ends of the system and guidewire, respectively. Manipulators A and C retracted the system in coordination. A balloon-expanded stent implantation was performed. A detailed operation video is presented in the Appendix A.

#### 3.1.4. Iliac Artery Stent Implantation

A 5 Fr pigtail catheter (Cook Inc.) was introduced along the guidewire into the arterial sheath. The surgeon subsequently positioned the four manipulators as follows: Manipulator A held the proximal part of the guide catheter. Manipulator B held the end of the 5 Fr catheter. Manipulators C and D held the proximal and distal guidewire, respectively. All four hands were coaxial. The catheter and guidewire were controlled using manipulators B and C, respectively. The guidewire was directed into the lower segment of the abdominal aorta and moved forward using the clamped manipulator C. Manipulator B pushed the catheter into the lower segment of the abdominal aorta in a coaxial direction. Subsequently, manipulator C withdrew its guidewire. The diameter of the target vessel (left common iliac artery) was measured using angiography. Subsequently, a 0.035 inch guidewire was inserted. Manipulators C and B coordinated to withdraw the pigtail catheter. This was followed by the introduction of a 5 Fr vertebral angiographic catheter (Terumo) along the guidewire. The guidewire was fixed using the manipulator B. Guided by the roadmap, manipulator C was rotated. The tip of the guidewire was directed towards the contralateral common iliac artery. It was pushed forward to the ultimate position of the third manipulator and then fixed. Subsequently, it was pushed forward by a second manipulator. Mediated by manipulators B and C, the guidewire was navigated to the contralateral external iliac artery. After retracting the vertebral angiographic catheter, an 8–60 mm self-expanding bare stent (Crownus, Microport) was introduced to the target position. The proximal part of the system was held by manipulator B. Additionally, manipulator D fixed the guidewire end. Manipulator C held the distal part of the system. Manipulator B retracted to release the stent. The stent system was reset by moving manipulators B and C in the opposite directions. Manipulators B and C subsequently retracted the system. The guidewire remained at rest. A relevant video is presented in the Appendix A.

#### 3.1.5. Implantation of Inferior Vena Cava Filter

Consistent with the aforementioned procedure, the right femoral vein was punctured to introduce the vascular sheath. The surgeon introduced a 5 Fr pigtail angiography catheter along the guidewire into the sheath. The four manipulators were positioned as follows: Manipulators A and B held the proximal and distal parts of the catheter, respectively. Manipulators C and D held the proximal and distal guidewire, respectively. All four hands were coaxial. An operation method similar to that used in Section 3.1 was employed. The 5 Fr pigtail catheter (Cook Inc.) was inserted into the inferior vena cava and removed at the end of angiography. Manipulators A/B grabbed the outer sheath of the filter system. They pushed it to the lower renal vein level and fixed it according to the roadmap. Manipulators C/D pushed the filter to the sheath tip. Manipulator D fixed the filter, whereas manipulator C clamped the outer sheath. It retracted, awaiting the complete release of the filter. Manipulator C clamped the inner core of the filter and rotated it continuously to release the filter. Finally, the inner core was retracted to the sheath to complete the process. A detailed operation video is presented in the Appendix A.

To evaluate the effect of the stent implantation and rule out any associated complications, final angiography was performed at the end of all procedures and after the exit of the stent system. Except for the vessel puncture, sheath placement, exchange of guidewire, and vessel sheath, all processes, including arterial navigation, femoral–femoral crossover, stent implantation, and filter release, were executed by the novel vascular interventional surgical robotic system. The total radiation exposure time was less than 1 min. Moreover, the operations were successful without complications, such as vessel dissection or perforation.

### 3.2. Outcomes

All operations were successfully completed. No adverse events were reported. Except for the expansion of the balloon-expanded stent, which was completed by the surgeon manually, the other actions were completed by the robotic system, including vessel navigation, endovascular devices delivery, stent deployment and withdrawal. Detailed information regarding the procedure outcomes is listed in Table 2. The angiograph results of five types of operations are shown in Figure 8.

### 3.3. Discussion

Manipulators for this novel vascular interventional surgical robotic system were primarily designed to satisfy the demands of gripping different devices. To realize guidewire navigation, catheter shaping, and stent release, a manipulator is required to rotate continuously. Adjusting the space between the double V-shaped grippers arbitrarily, as per the requirements of a universal robotic system capable of performing all types of endovascular surgery, ensures a full-size fit of the 0.014 inch guidewire and aortic stent graft system. In addition, the manipulator is expected to clamp the treatment instrument without displacement and expand the adaptable range. In experiments on animals using Corpath (Corindus Vascular Robotics, Inc., Waltham, MA, USA), the guidewire was moved forward with a catheter [18]. Notably, if this occurs during the coil embolization of intracranial aneurysms, catastrophic results, including aneurysm hemorrhage or vascular perforation, will arise. During a manual operation, the surgeon or assistant holds the guidewire to curtail its forward movement. Corpath adopts a roller-stepping technique, shifting the guidewire when the friction between the guidewire and the catheter decreases. Corpath GRX (Corindus Vascular Robotics, Inc.) has a fail-safe mechanism that can avoid this scenario. However, this decrease in the catheter is attributed to the inconsistent mechanical designs that involve manual operations [19]. This explains why the novel vascular interventional surgical robotic system employs a double V-shaped gripper to simulate manual gripping. It minimizes the additional risks associated with applying a new mechanical design to the existing endovascular devices. The frictional force between the catheter and manipulator is a crucial technical parameter. It depends on the clamping motor power, catheter material, and material of the clamping device. During the entire operation, the demand for friction varies in most cases. However, the minimum friction requirement should be met to reduce device damage. The maximum clamping friction of the new double V-shaped gripper of the robot is more than 13 N. It shows good compatibility with the currently available guidewire, catheter, and stent system without displacement. Additionally, compared to the rolling friction technology applied in the Corpath system, the V-shaped gripper causes less damage to the hydrophilic coating of endovascular devices. It considerably reduces the risk of vascular wall damage and better regulates the medical costs. Moreover, for stable axial velocity, the manipulator must be controlled using a PID controller. The expected speed of the manipulator was found to remain unchanged with a PID controller, when disturbed with a short response time.

In our study, to minimize the risk of entrapping the guidewire with the manipulator in continuous rotation, we adopted the progressive rotation method for the two groups of grippers. The double V-shaped grippers rotated 90° each time. The third V-shaped relay gripper was relayed to maintain the rotation angle until it was reset and rotated again. The rotation process was automated using manipulators. The surgeon rotated the joystick to issue rotation instructions and released it after achieving the desired angle. The rotation accuracy of the autonomous rotation mode reached 1°. The precision, that is, the accuracy of prediction, was predictable. It met the interventional surgery requirements. The guidewire and catheter used by RobEnt were fixed in the gear. The rotation of the guidewire and catheter was powered by gear rotation [12]. It increased rotational speed. However, the replacement is difficult. The average loading time was 100 s, whereas the disassembly time was 30 s. For the novel vascular interventional surgical robotic system, device replacement was controlled within 10 s, which shortened the operation time.

Another drawback in the application of vascular interventional surgical robotic systems is that a considerable proportion of the current clinical operations are performed by surgeons and assistants. The operating path between the operator and assistant alternates between the coaxial or non-coaxial states depending on the device type. To simulate such complex movements, we extended the freedom of movement of four manipulators. The 12-DOF motion in the four directions of the short axis were executed independently via the coordinated movement of the manipulators. Consequently, real-time transformation between the coaxial and non-coaxial states was achieved. The axial state of the manipulators was controlled using software. The operating joystick controls the forward, backward, and rotational movements of the manipulator. Thus, accidental changes in the axial angle can be avoided during surgery. The superposition of torque in the direction of the axial deviation angle can be prevented. The device might damage the endovascular device and increase the risk of iatrogenic injury. The novel vascular interventional surgical robotic system can execute relatively complex operations, including crossover, double-guidewire navigation, and self-expanding stent implantation. It overcomes the technical defects of the current endovascular interventional robots [9,10]. In addition, it can complete catheter advancement and exchange by presetting automatic operation steps, particularly in debulking operations including thrombolytic drug spraying and thrombectomy. In short, the automatic standard operation of the endovascular robotic system can avoid the impact of inexperienced surgeons on the therapeutic effect. It is beneficial to promote the application of new technology.

Studies have demonstrated that guidewire-related and catheter-related iatrogenic dissection during endovascular treatment occurs in approximately 3.6% of cases [20]. The risk range for bleeding complications is 0.7% and 1.0%. The joystick can execute all or none of the operational instructions at a constant speed of the horizontal movement and rotation. Notably, the advancing speed was adjusted to exert a certain degree of rotation when the guidewire and catheter passed through narrow or distorted lesions. Movement adjustment was based on visual feedback. However, for clinical requirements, particularly in occlusive lesions, where visual feedback cannot be provided by vessel wall deformation, retaining tactile feedback is crucial for a successful treatment.

Sterility is directly related to the success of the operation. Incomplete sterilization increases the risk of infection. Corpath GRX and R-One adopt disposable cassettes to address the sterilization problem, but this usually comes at a high cost. Therefore, sterilization with high efficiency and low cost is a critical challenge in the clinical application of robots. To address this challenge, the detachable design of the manipulator gripper approaches the replacement of all contact parts with the endovascular device during pushing, retreating, and rotating. One operation using a one-person approach was accomplished. This reduced the risk of cross-infection and medical costs.

## 4. Conclusions

In this study, we optimized the operational logic of the joystick of a novel vascular interventional surgical robotic system. We established a set of handles based on the feedback mechanism of electromagnetic resistance, retaining the logic of manual operation to the greatest extent possible. Radiation exposure time of surgeons can be shortened significantly. The relatively larger movements on the handle generated by the surgeon can be transformed into subtle movements on the device. This improves sensitivity and stability and effectively reduces the risk of iatrogenic injury. Moreover, surgeons can select the most comfortable mode of operation to further reduce operative fatigue.

## Figures and Tables

**Figure 1 micromachines-13-01587-f001:**
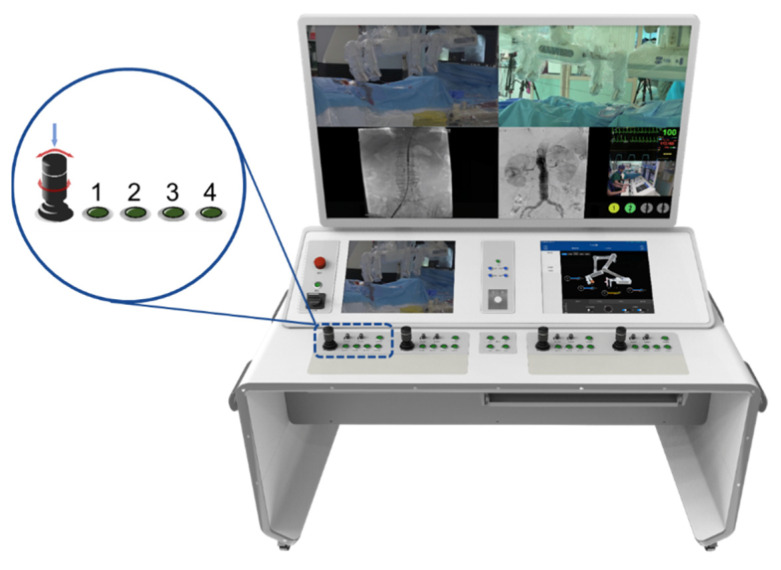
An overview of the interventional console and the model of the joystick in the interventional console. The upper part of the joystick is a button and can be pressed to control the loosening and clamping of manipulators. The lower part of the joystick is a knob and can be rotated to control the rotation of manipulators. Left and right swing of the joystick can control the advancement and retraction of manipulators. Control of manipulators can be chosen by pressing different green selection buttons (one bottom refer to one manipulator).

**Figure 2 micromachines-13-01587-f002:**
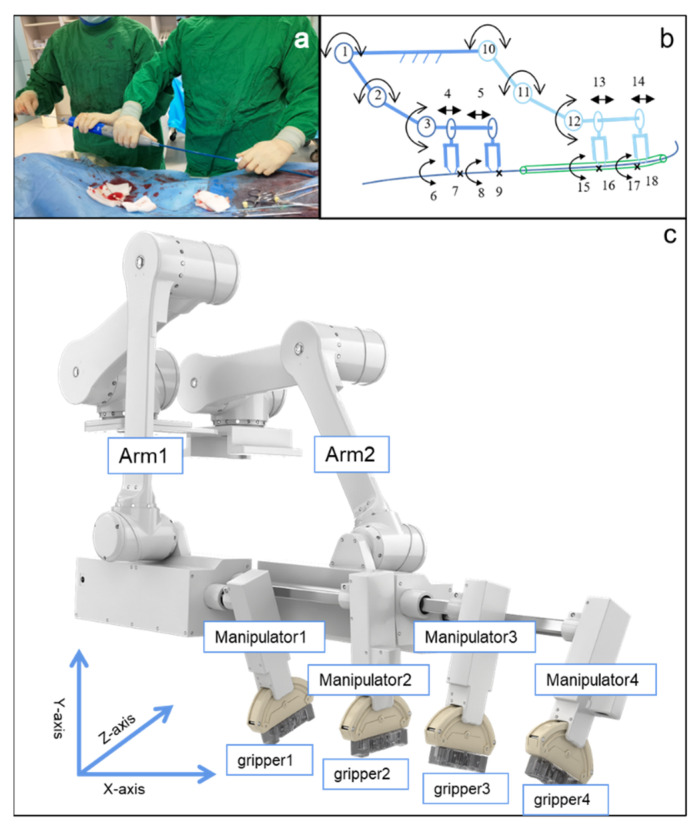
Bionic design of endovascular robotic system (**a**): the manual operation; (**b**): the illustration of bionic design; (**c**): Arm 1 and Arm 2 simulate the operator and assistant which can be moved in all the X, Y, and *Z*-axis directions. The manipulators simulate the hands of the operator which can be moved in the *X*-axis direction. The grippers simulate the fingers of the operator which can be rotated around the *Z*-axis plane.

**Figure 3 micromachines-13-01587-f003:**
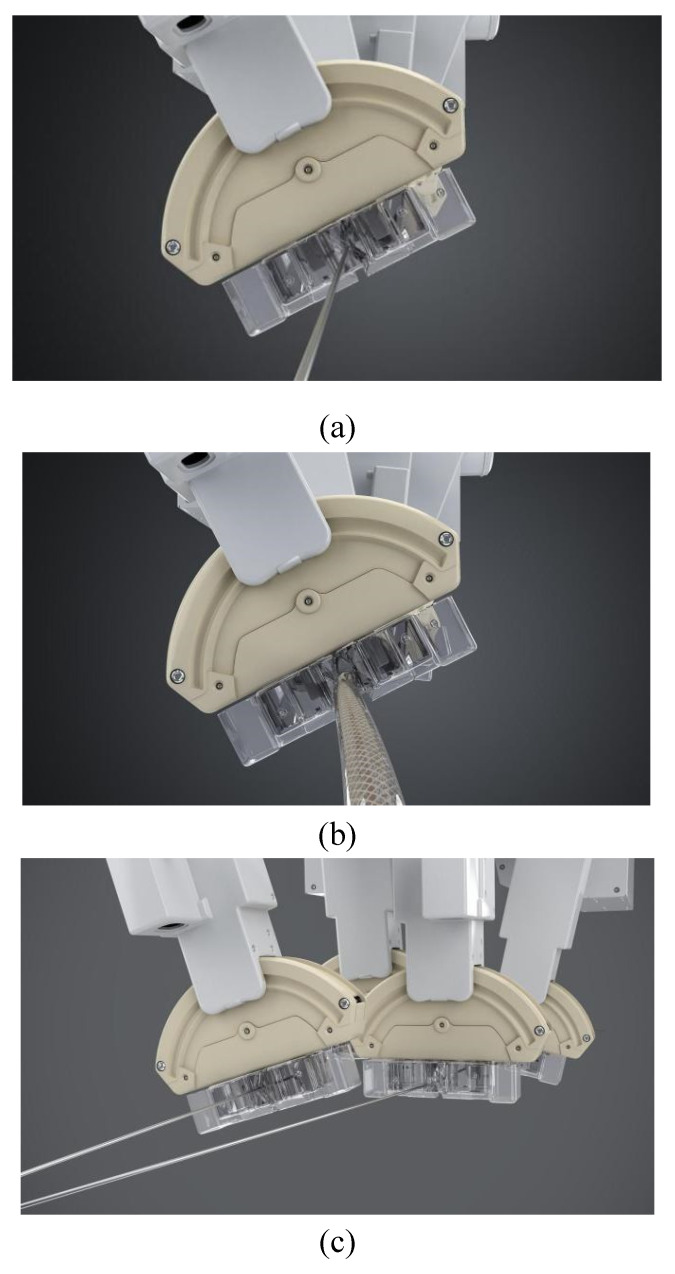
(**a**) Coaxial state of four manipulators (guidewire); (**b**) Coaxial state of four manipulators (stent graft); (**c**) Non-coaxial state of four manipulators (rapid exchange system).

**Figure 4 micromachines-13-01587-f004:**
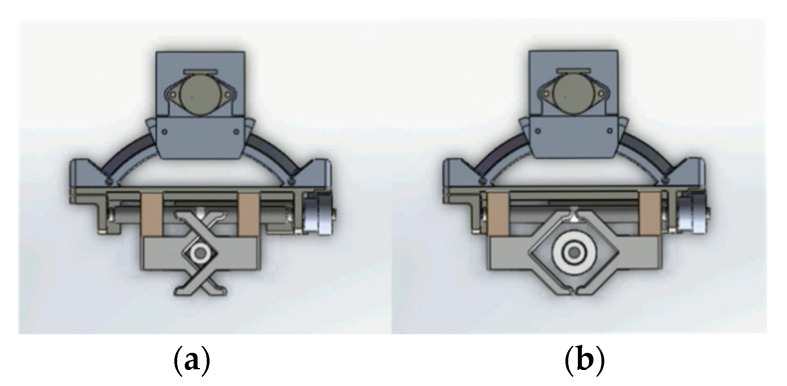
The working states of the double V-shaped grippers (**a**): clamping state; (**b**): loosening state.

**Figure 5 micromachines-13-01587-f005:**
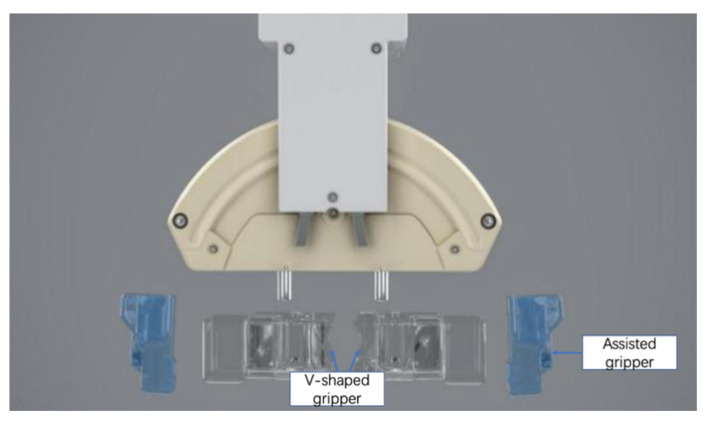
Detachable design of the grippers.

**Figure 6 micromachines-13-01587-f006:**
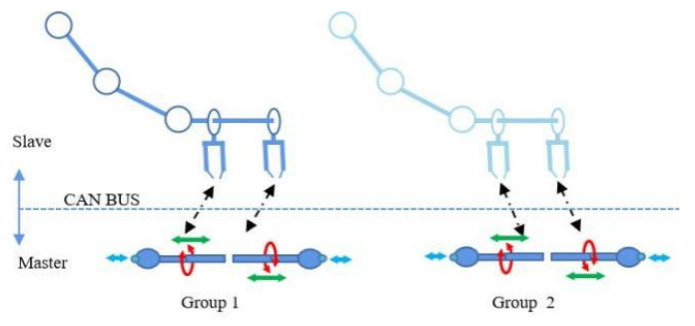
Master–slave operation design. The green arrows indicate that when the joystick is pulled or pushed, the slave manipulator advances or retracts. The blue arrows indicate that when the operator presses the button at the top of the joystick, the gripper clamps or releases. The red circle means that when the operator rotates the joystick, the gripper rotates.

**Figure 7 micromachines-13-01587-f007:**
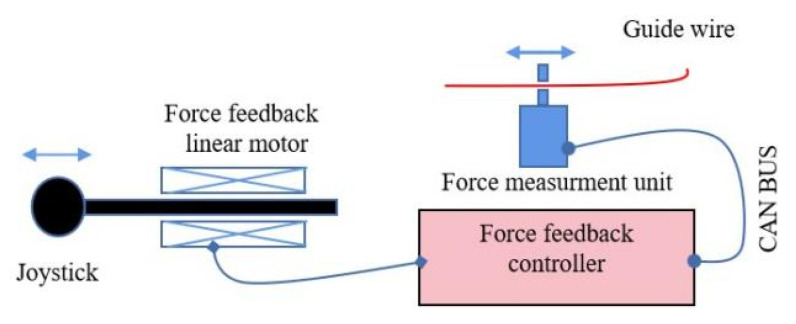
The illustration of force feedback handle.

**Figure 8 micromachines-13-01587-f008:**
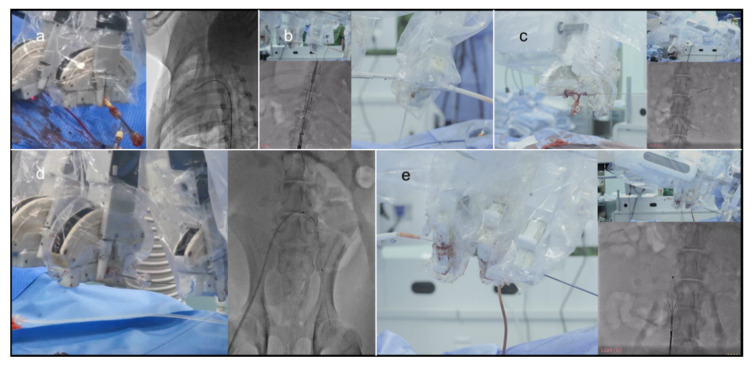
The angiograph results of five types of operations. (**a**): Percutaneous coronary intervention; (**b**): thoracic endovascular aortic repair; (**c**): stent implantation of renal artery; (**d**): iliac artery stent implantation; (**e**): implantation of inferior vena cava filter.

**Table 1 micromachines-13-01587-t001:** Basic performance parameters of the robot.

Performance Parameter	Value
maximum axial motion speed	275 mm/s
minimum axial motion speed	5.15 mm/s
maximum propulsion power	7.55 W
maximum thrust	27.45 N
maximum displacement distance	140 cm

**Table 2 micromachines-13-01587-t002:** Detailed information regarding the procedure outcomes.

Items	Percutaneous Coronary Intervention		*p* Value	Thoracic Endovascular Aortic Repair		*p* Value	Stent Implantation of Renal Artery		*p* Value	Iliac Artery Stent Implantation		*p* Value	Implantation of Inferior Vena Cava Filter		*p* Value
	Robotic	Manual		Robotic	Manual		Robotic	Manual		Robotic	Manual		Robotic	Manual	
**Anesthesia**	Local			General			Local			Local			Local		
**Stents**	Firebird (Microport)			Hercules (Microport)			Firebird (Microport)			Everflex (EV3)			Everflex (EV3)		
**Number of animals**	8	8		4	4		8	8		10	10		8	8	
**Average operative time (min)**	42.5 ± 5.13	40.4 ± 6.57	0.488	80.4 ± 8.01	75.3 ± 5.81	0.342	40.4 ± 3.11	39.5 ± 5.31	0.685	43.8 ± 6.51	39.8 ± 5.23	0.147	25.0 ± 2.50	24.9 ± 2.85	0.942
**Blood loss (ml)**	50.9 ± 4.48	51.6 ± 5.99	0.795	74.3 ± 1.92	74 ± 5.52	0.922	42.3 ± 6.81	44.1 ± 10.30	0.686	47.6 ± 4.47	45.1 ± 8.2	0.408	34.3 ± 1.98	35.4 ± 2.82	0.382
**Total contrast volume used (ml)**	148.1 ± 2.58	146.9 ± 2.59	0.369	205.6 ± 4.96	206.3 ± 10.60	0.909	146.9 ± 3.73	145.6 ± 4.17	0.521	175.0 ± 3.78	179.4 ± 4.95	0.038	102.5 ± 4.63	105.6 ± 4.17	0.181
**Fluoroscopy time (min)**	39.9 ± 5.48	36.8 ± 2.19	0.160	65.3 ± 4.22	58.3 ± 4.36	0.06	36.4 ± 3.97	32.5 ± 4.14	0.075	39.8 ± 4.91	35.8 ± 5.12	0.091	20.0 ± 3.35	19.5 ± 2.91	0.755
**Procedural radiation exposure Operator (μGy)**	24.6 ± 2.72	2224 ± 92.5	<0.001	35.3 ± 1.71	2204 ± 70.64	<0.001	24.8 ± 2.81	2453.5 ± 207.43	<0.001	23.3 ± 2.49	2293 ± 154.63	<0.001	16.1 ± 3.27	1388 ± 252.46	<0.001
**Procedural radiation exposure Patient (mGy)**	2239.6 ± 109.79	2224 ± 92.5	0.763	2002 ± 241.71	2204 ± 170.64	0.221	2338.6 ± 107.45	2453.5 ± 207.43	0.186	2410.5 ± 208.62	2293 ± 154.63	0.17	1551 ± 225.70	1388 ± 252.46	0.195
**Technical success**	Y	Y		Y	Y		Y	Y		Y	Y		Y	Y	
**Perioperative death**	-	-		-	-		-	-		-	-		-	-	
**Perioperative complications**	-	-		-	-		-	-		-	-		-	-	

## Data Availability

The data presented in this study are available on request from the corresponding author.

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
