# Peer review of "Novel Endovascular Interventional Surgical Robotic System Based on Biomimetic Manipulation"

_micromachines, 2022, doi:10.3390/mi13101587_

Round 1
Reviewer 1 Report
The main contribution of this paper is the development of a novel surgical robotic system for endovascular intervention.
The paper has enough work with the contribution of a novel robotic system and its functional tests. But the paper needs to be improved significantly. The structure of the paper is not very clear, and some concrete contents should be added. The design, assessment and experiment are all mixed together; the detail structure description and working principle of the system are missing. The assessment is more like demonstration and description of using the system without quantitative result.
There are some detailed comments below:
Materials and method
The working principle, structure, system control of the novel device and how those design are superior than the other systems need to be written clearly.
· As the paper mainly claim the novel device, I think it may be clearer to have separate subsection(s) to introduce the working principle, structure, and control of the new robotic system and then assess each function.
· The paper emphasizes this novel device is based on biomimetic manipulation. Please state clearly how does this device based on biomimetic manipulation and follow bionic design and why this design is better.
· Please describe the system structure in detail, e.g., mark the components and show inner structure in Fig1c, Fig 3&4.
Results and Discussion
· Is there any criterion to show the performance of the operation quantitively, e.g., operation time, accuracy etc? The current content is more like demonstration and detailed description of the procedure, but it is not known whether the operation is good or not.
· Some content on experiment detail, like how many trials has been conducted, how many surgeons has joined and their experience on surgery and using this new system, whether they need training or practice before using the system, etc., need to be added.
· Videos attached are helpful and it is good to see the comparison of this system in the discussion of section 3.7.
· It will be good to add some quantitative result comparisons of this new system with the manual procedure or other robotic system in section 3.2-3.6 to clearly show the effectiveness of the proposed system in different surgical procedure.
Reviewer 2 Report
See attached

Round 2
Reviewer 1 Report
Thank you for the reply. My comments have been addressed by the authors. Some minor comments on the new added table 2, the front is small to read and for result of robotic and manual controls in the table, is there any statistical difference between them?
Author Response
Thank you for your positive comments and suggestions. We have performed the statistical analysis for the result of robotic & manual controls in the table 2, and enlarged the font and the size of the table 2 as you can see in the manuscript. The table 2 has also been included in the supplementary materials.
Reviewer 2 Report
The authors have answered my questions.
Author Response
Thank you for your positive comments.